# Nano- And Microfiber-Based Fully Fabric Triboelectric Nanogenerator For Wearable Devices

**DOI:** 10.3390/polym12030658

**Published:** 2020-03-13

**Authors:** Jong Hyuk Bae, Hyun Ju Oh, Jinkyu Song, Do Kun Kim, Byeong Jin Yeang, Jae Hoon Ko, Seong Hun Kim, Woosung Lee, Seung Ju Lim

**Affiliations:** 1Smart Textiles R&D Group, Korea Institute of Industrial Technology (KITECH), Ansan 31056, Korea; baejh@kitech.re.kr (J.H.B.); songjk543@kitech.re.kr (J.S.); ellafiz@kitech.re.kr (J.H.K.); 2Department of Organic and Nano Engineering, Hanyang University, Seoul 04763, Korea; 3Technical Textile R&D Group, Korea Institute of Industrial Technology (KITECH), Ansan 31056, Korea; hjoh33@kitech.re.kr (H.J.O.); dogun419@kitech.re.kr (D.K.K.); yeang777@kitech.re.kr (B.J.Y.); 4Department of Advanced Materials Engineering for Information & Electronics, Kyung Hee University, Yongin 17104, Korea

**Keywords:** triboelectric effect, triboelectric nanogenerator, nonwoven nanogenerator, nylon nanofiber, polypropylene microfiber

## Abstract

The combination of the triboelectric effect and static electricity as a triboelectric nanogenerator (TENG) has been extensively studied. TENGs using nanofibers have advantages such as high surface roughness, porous structure, and ease of production by electrospinning; however, their shortcomings include high-cost, limited yield, and poor mechanical properties. Microfibers are produced on mass scale at low cost; they are solvent-free, their thickness can be easily controlled, and they have relatively better mechanical properties than nanofiber webs. Herein, a nano- and micro-fiber-based TENG (NMF-TENG) was fabricated using a nylon 6 nanofiber mat and melt blown nonwoven polypropylene (PP) as triboelectric layers. Hence, the advantages of nanofibers and microfibers are maintained and mutually complemented. The NMF-TENG was manufactured by electrospinning nylon 6 on the nonwoven PP, and then attaching Ni coated fabric electrodes on the top and bottom of the triboelectric layers. The morphology, porosity, pore size distribution, and fiber diameters of the triboelectric layers were investigated. The triboelectric output performances were confirmed by controlling the pressure area and basis weight of the nonwoven PP. This study proposes a low-cost fabrication process of NMF-TENGs with high air-permeability, durability, and productivity, which makes them applicable to a variety of wearable electronics.

## 1. Introduction

Fiber-based electronic devices are attracting extraordinary attention due to their flexibility, lightness, comfortableness, and applicability to a variety of industries and products, including physical or chemical sensors [1,2], biomedical monitoring [3,4], soft robotics [5,6], and wearable devices [7,8]. Furthermore, various portable smart devices are playing major roles in daily life, and these devices require lighter, smaller, or larger capacity power sources. Related research topics such as sensors [9,10], energy harvesting [11,12,13,14], piezoelectric nanogenerators (PENGs) [15,16,17], and triboelectric nanogenerators (TENGs) have received significant attention [18,19,20,21,22,23] because a large amount of available energy is generated by the human movement and clothing friction. The triboelectric effect is caused by the contact between different materials, and it can induce strong electrostatic charges. As static electricity can lead to ignition, dust explosions, and electrical shocks, the unintended static electricity is generally considered to have a negative impact in the industry. However, the combination of the triboelectric effect and static electricity has been extensively studied as a TENG [24,25,26,27,28] because it produces a sufficiently large amount of electrical energy to be used as a generator. An all-nanofiber-based stretchable TENG (S-TENG) with polyvinylidene fluoride (PVDF) and thermoplastic polyurethane (TPU) nanofiber membranes was reported by Zhao et al. [29] for energy harvesting. This S-TENG, which according to the analysis had a full separation of surface-to-surface, had an excellent triboelectric output performance. Zhu et al. [30] introduced the microfiber-based TENG in 2018. This TENG consisted of ZnO-coated polypropylene (PP) microfibers with a spacer, and it exhibited high transfer charge and output voltage. The properties of a TENG depend on various elements, such as the surface morphology, dielectric constant, spacer, and triboelectric potential difference between the triboelectric materials [31,32,33,34].

Since the fiber-based TENG was introduced by Zhong et al. in 2014 [35], related studies have been actively reported [15,16,36]. Notably, the TENG using nanofibers has been widely investigated because of its advantages, such as ease of production by electrospinning and high surface roughness [37,38,39,40,41]. Owing to the porous structure of nanofibers, which can contain a large volume of air with high dielectric constant, they have a large contact area, which can enhance the triboelectric effects and produce a high-output generator [42,43,44]. However, TENGs are generally designed with structures that include spacers to enhance the electrical performance, and this leads to increased fabrication process steps, cost, and total volume of devices. In addition, nanofibers can be produced only by electrospinning with high-cost and limited yield, which also results in relatively low mechanical properties due to the difficulty in improving strength through the orientation of polymers [45].

The melt blowing process is a well-known method of producing nonwoven fabrics and can be applied to various thermoplastics, including polyethylene terephthalate (PET) [46], polyolefin [47], and polylactic acid (PLA) [48]. Melt blown nonwoven fabrics, which generally consist of microfibers, can be produced with low cost on a mass scale in large areas, are solvent-free, their thickness is easily controlled, and have relatively better mechanical properties than nanofiber webs [49,50].

In this study, in order to maintain and mutually complement the advantages of nanofibers and microfibers and avoid their shortcomings, a nano- and micro-fiber-based TENG (NMF-TENG) was fabricated using a polyamide (PA), especially nylon 6, nanofiber mat and melt blown nonwoven polypropylene (PP) as triboelectric layers. The NMF-TENG is composed of the nylon 6 solution electrospun on the nonwoven PP and Ni-coated fabric electrodes. The nanofiber mat and nonwoven PP contain a large volume of air in the porous structure, and thus, the NMF-TENG does not require a spacer. The morphology, porosity, pore size distribution, and fiber diameters of the triboelectric layers were characterized. Moreover, the electrical output performances of NMF-TENGs were investigated. This study proposes a low-cost fabrication process of NMF-TENGs with high triboelectric output performance, air-permeability, durability, and productivity. Therefore, their application to a variety of wearable electronics is expected.

## 2. Experimental Section

### 2.1. Materials

A PP pellet (HP561X, melt-blown grade, Polymirae, Korea) with 800 g/10 min of melt flow rate (MFR) and 0.9 g/cm^3^ of density was used as a polymer for the melt-blowing process. Nylon 6 (1011 BRT) was received from Hyosung (Korea). Formic acid and acetic acid as solvents were purchased from Samchun (Korea).

### 2.2. Fabrication of Nonwoven Triboelectric Layers

The melt blowing of PP was performed with a pilot scale melt blown line (KIT MB 2005, Hills Inc., Melbourne, FL, USA). The PP pellet was melted at 250 °C in the extruder and extruded through the spinneret. The air gap was 0.4 mm, and the nozzle to collector distance was 300 mm. The air temperature was set at 260 °C. The fabricated melt blown nonwoven basis weight was varied between 15, 30, and 50 gsm.

To fabricate the nanofiber mat, the nylon 6 was dissolved in the mixture solvent of formic acid and acetic acid (8:2) in a concentration of 15 wt%. The dissolved solution was ejected on the obtained nonwoven PP through the metal nozzle (25G) connected to a high-voltage generator (NNC-HV30, NanoNC, Seoul, Korea). The applied voltage and feed rate in the solution were 15 kV and 10 uL/min, respectively. The distance from the metal nozzle to the collector was 15 cm. After that, the prepared nanofiber mat was dried at 50 °C under vacuum for 24 h.

### 2.3. Fabrication of the NMF-TENGs

The NMF-TENGs were fabricated based on the vertical contact mode among the fundamental operation modes. The NMF-TENG contains two parts: the triboelectric layer with nonwoven PP (161–475 μm thick) and nylon 6 nanofiber mat (80 μm thick), and a Ni-coated conductive fabric (110 μm thick), as the top and bottom electrodes. The triboelectric layer was cut into 50 mm × 50 mm, and the Ni fabric electrode was cut to 45 mm × 65 mm. The Ni fabric electrodes were attached to the top and bottom of the triboelectric layer. Finally, the NMF-TENGs were easily fabricated without spacers.

### 2.4. Characterization

The surface images of the nonwoven PP and nylon 6 nanofiber mat were observed by field emission scanning electron microscopy (FE-SEM) (SU8010, Hitachi Co., Tokyo, Japan) with an acceleration voltage of 10 kV after sputter coating with osmium (Os). The pore size distribution of these specimens was measured by a capillary flow porometer (CFP-1500-AEX, PMI Inc., Ithaca, NY, USA) according to the ASTM F316-03 standard. The specimen size was 20 mm × 20 mm, and it was obtained by following equation:d = Cγ/p,(1)
where d is the pore diameter (μm), γ is the surface tension (mN/m), p is the pressure (Pa), and C is a constant (2860) [51]. The porosity of the nonwoven PP and nylon 6 nanofiber mat were characterized by a mercury porosimeter (Autopore IV 9500, Micromeritics, Norcross, GA, USA). To analyze the electrical output performance of the NMF-TENGs, the open-circuit voltage (V_oc_) and the short-circuit current (I_sc_) were measured and recorded by a digital oscilloscope (Keysight, DSOX4024A, Santa Rosa, CA, USA) and a source meter (Keysight, B2901A, Santa Rosa, CA, USA), respectively.

## 3. Results and Discussion

The manufacturing process of the NMF-TENG is shown in Figure 1a. In the first step, nonwoven PP was manufactured in the pilot scale equipment. After that, the nanofiber mats were electrospun on the nonwoven PP fabric, and an integrated NMF-TENG without a spacer was fabricated. As shown in Figure 1b, the nonwoven PP was randomly distributed. In the case of the nanofibers, the structure of the fabricated mats was more dense and compact than that of the nonwoven PP. Figure 1c shows the chemical structure both of PP and nylon 6, and it also presents the cross-sectional FE-SEM image of the NMF-TENG specimen. Although it is combined without a spacer between the two triboelectric materials, there is no full contact between the rough fibers owing to the porous structure of the nonwoven fabric.

The structural characteristics of the nonwoven PP and nylon 6 nanofiber mat are summarized in Figure 2 and Table 1. In the case of the nonwoven PP, its average diameter was in the range of 2.4–2.7 μm as the basis weight of the nonwoven PP increased. The thickness of the nonwoven PP increased up to approximately two times with the increasing weight. The air permeability decreased with increasing weight, and it is assumed that the air pathway increases in the thickness direction. In Figure 2a, the pore size distribution of the PP membrane with the various basis weights was exhibited. The pore size showed a relatively broad range, which was not affected by the increase in basis weight of the nonwoven PP. The average and maximum pore sizes were 16 and 33 μm. The porosity of the nonwoven PP was slightly increased, up to 80%, as the basis weight of the nonwoven PP increased. It was confirmed that the air permeability and porosity of the nonwoven PP is higher than that of other film-based materials commonly used in TENGs. In addition, the pore area of the PP50 basis weight related to the air volume was significantly increased, and the area of fiber contact between the materials could also be increased with an applied force. For the nylon 6 nanofiber mat, the fiber diameter was 283 nm, which was ten times smaller than that of the nonwoven PP. The pore size was sharply distributed around 320 nm, and the maximum pore size was 800 nm. From the previous FE-SEM image, the structure of the nylon 6 nanofiber mat was relatively denser than that of the nonwoven PP. The porosity of the nanofiber mat was approximately 87%, a high pore volume. From those results, we confirmed that both the nonwoven PP and nylon 6 nanofiber mats have a highly porous and open pore structure through the thickness direction.

A schematic of the NMF-TENG under the vertical contact mode is shown in Figure 3a. The nylon 6 nanofiber mat and nonwoven PP are chosen as the negative and positive triboelectric layers, respectively. The nylon 6 nanofiber mat is successfully deposited by electrospinning onto the nonwoven PP surface, and the Ni fabric electrodes are attached to the top and bottom of the triboelectric layers. Figure 3b shows a schematic of the NMF-TENG to which an external force is applied and released. The triboelectric layers of the NMF-TENG contain numerous pores with a volume of air and critical contact points between the fibers. As the external force with vertical components is applied, the triboelectric layers are deformed, which results in a decrease in the air volume and an increase in critical contact points between the frictional materials. Therefore, by applying a mechanical force such as pressing and releasing, surface triboelectric charges are generated due to changes in the contact area between frictional materials having different polarities. Figure 3c shows a schematic of the electricity generation mechanism in the vertical contact mode. In the original state, there is no generation of frictional charges or potential differences between the two electrodes [52]. When an external force is applied to the top surface of the NMF-TENG, the triboelectric layer is deformed as described above, and then triboelectric charges are generated by the change in contact area between the nylon 6 nanofibers and PP microfibers. When the external force is released, the opposite charges are separated, and electrons flow through the external circuit, inducing the potential difference. As a result of this sequence, the open-circuit voltage and short-circuit current shown in Figure 3d are generated during one cycle by the external force applied to the NMF-TENG surface.

The effect of the PP basis weight on the electrical performance of the NMF-TENGs was investigated by applying an external force to NMF-TENGs with different PP basis weights. The fabricated NMF-TENGs were periodically pressured and released under the constant force of 5 N at a frequency of 8 Hz.

The NMF-TENGs generated excellent output voltage and current, as shown in Figure 4, despite the absence of a spacer in the structure. The electrical output performance of PP15 shows V_oc_ of 1.55 ± 0.12 V and I_sc_ of 161.01 ± 10.12 nA. The electrical performance of PP30 shows V_oc_ of 2.79 ± 0.13 V and I_sc_ of 243.88 ± 12.42 nA, which is an improved output performance over that of PP15. Furthermore, when the PP basis weight is increased to 50 gsm, V_oc_ and I_sc_ are further improved to 3.54 ± 0.13 V and 374.39 ± 22.28 nA, respectively. Based on these results, the electrical output performance of the NMF-TENG was improved with an increase in the PP basis weight from 15 to 50 gsm. These results arise because the high basis weights of nonwoven PP contain a high porosity and large volume of air with a high dielectric constant, thereby enhancing the triboelectric effect.

In this study, the NMF-TENG fabricated by the use of nonwoven PP50 and a nylon 6 nanofiber mat is selected to examine the triboelectric properties. The electrical output performances of the fabricated NMF-TENG were investigated by applying various external forces. The NMF-TENG was pressured and released for three cycles with increasing force to 0.5, 2, and 5 N in a mechanical pressure machine (SnM Tech, Korea), as shown in Figure 5a. Figure 5b illustrates the electrical response of the NMF-TENG in terms of voltage as the force increases. As shown in the results, it can be observed that the output voltage of the NMF-TENG increases to 1.26 ± 0.01, 2.38 ± 0.24, and 3.47 ± 0.06 V as the pressure varies between 0.5, 2, and 5 N, respectively. Figure 5c shows the electrical response of the NMF-TENG in terms of current. The results indicate that the output current increases to 107.76 ± 7.07, 150.23 ± 9.71, and 247.58 ± 10.71 nA as the pressure gradually increases. As mentioned above, the increase in electrical performance of the NMF-TENG in response to an increasing pressure is related to a change in the interfacial contact area between the friction materials.

Moreover, the electrical charge potential of the NMF-TENG was realized by finger (Figure 5d), blade (Figure 5e), and palm tapping (Figure 5f) with the human’s hand, and the generated output voltages were 12.52 ± 0.87, 23.10 ± 0.60, and 33.93 ± 2.09 V, respectively, despite the application of a similar pressure, from approximately 14 to 16 N. The results indicate that the area of pressure applied to the surface of the NMF-TENG is related to the output performance. It means that the charge potential of the inner friction area is increased through the increase in pressure area. To demonstrate the capability of the NMF-TENG for energy harvesting, it was used as a power source to charge capacitors through a bridge rectifier under a periodic external force of 5 N at 8 Hz, as shown in Figure 5g. As plotted in Figure 5h, the capacitor with 0.1 μF is rapidly and steeply charged by the NMF-TENG. However, as the capacitance becomes larger, a characteristic linear charging behavior occurred. Furthermore, we replaced the capacitor with LEDs in the bridge rectifier, and only the power source generated by the NMF-TENG was used to turn on the LEDs, as shown in Figure 5i. The full fabric-type NMF-TENG can be considered as a power source for wearable devices.

## 4. Conclusions

In summary, we have developed a nano- and micro-fiber-based TENG (NMF-TENG) by using a nylon 6 nanofiber mat and melt blown nonwoven PP as the triboelectric layers to enhance the electrical performance. The NMF-TENG was fabricated by electrospinning nylon 6 on the nonwoven PP with Ni coated fabric electrodes. Owing to the porous structure of the triboelectric layers containing a large volume of air with a difference in electronegativity, our NMF-TENG could generate triboelectric effects even with very thin layers and without a spacer. The morphology, porosity, pore size distribution, and fiber diameters of the triboelectric layers were characterized. Moreover, the electrical output performance of NMF-TENGs was investigated. When PP15, PP30, and PP50 were applied to the triboelectric layer with a nylon 6 nanofiber mat, the electrical output performance of the NMF-TENG was improved with the increase in PP basis weight from 15 to 50 gsm. Furthermore, the electrical charge potential realized by hand tapping with different pressure areas was 33.93 V for palm, 23.10 V for blade, and 12.52 V for finger tapping. It means that the charge potential of the inner friction area is increased through the increase in pressure area. To demonstrate the capability of the NMF-TENG for energy harvesting, it was used as a power source to turn on LEDs.

The triboelectric output performance of NMF-TENGs with high permeability provides the opportunity to apply them to smart and wearable device power systems and self-generated electronic systems. The large area of the NMF-TENG and its application to clothing will be studied in the near future.

## Figures and Tables

**Figure 1 polymers-12-00658-f001:**
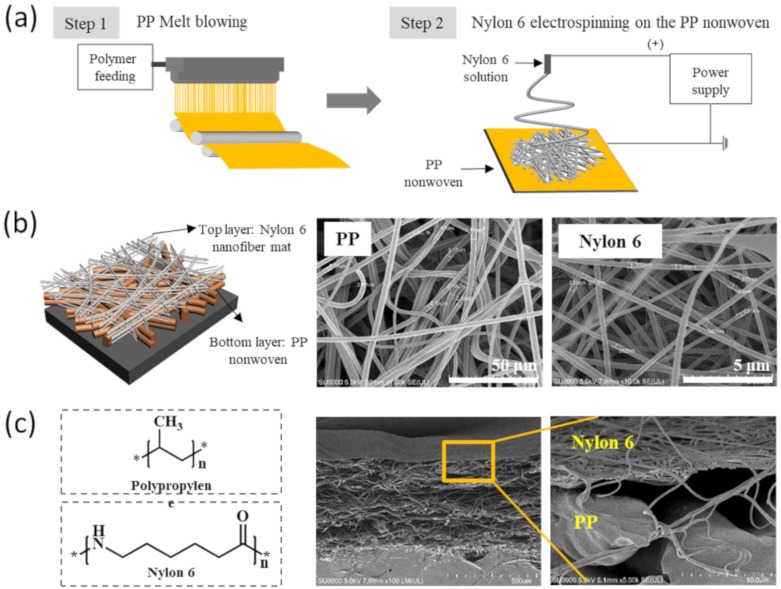
(**a**) Schematic diagram of the fabrication process and surface morphology of the nonwoven polypropylene (PP) and nylon 6 nanofiber mat, (**b**) the fabricated nano- and micro-fiber-based triboelectric nanogenerators (NMF-TENGs) structure with field emission scanning electron microscopy (FE-SEM), (**c**) chemical structures of the PP and nylon 6, and the cross-sectional FE-SEM images of the nonwoven PP layered with the nylon 6 nanofiber mat.

**Figure 2 polymers-12-00658-f002:**
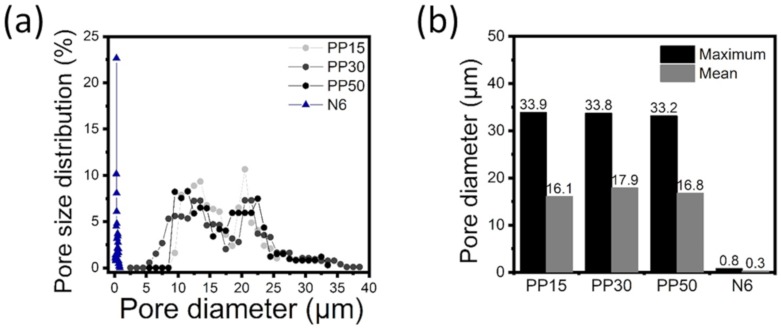
(**a**) Pore size distribution, and (**b**) maximum and average pore diameter of the nonwoven PP according to the basis weight and the nylon 6 nanofiber mat.

**Figure 3 polymers-12-00658-f003:**
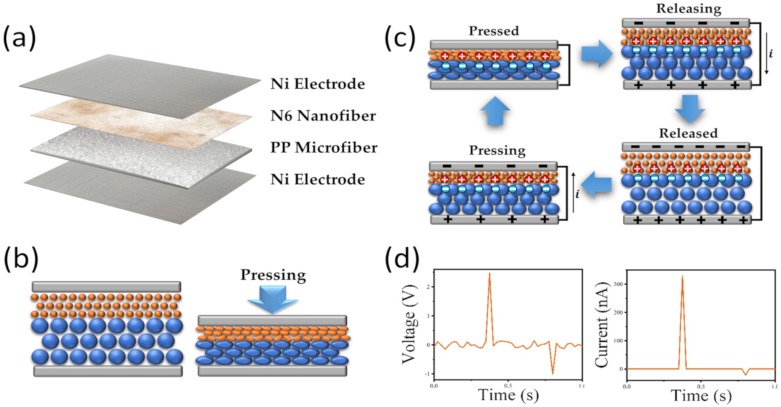
(**a**) Schematic of NMF-TENGs, (**b**) schematic of NMF-TENGs when external forces are applied and released, (**c**) schematic of the working mechanism of the triboelectric nanogenerators, and (**d**) electric output performances of NMF-TENGs generated by an external force during one cycle.

**Figure 4 polymers-12-00658-f004:**
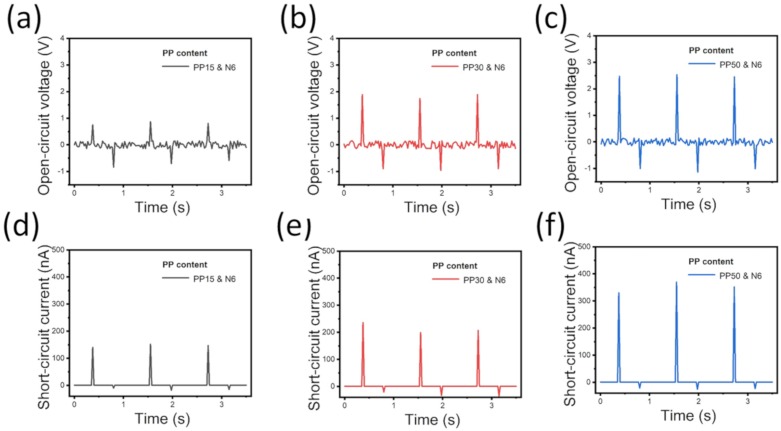
Effect of PP basis weight on the electrical output performance of the NMF-TENGs. Output voltage of NMF-TENG with (**a**) PP15, (**b**) PP30, and (**c**) PP50. Output current of NMF-TENG with (**d**) PP15, (**e**) PP30, and (**f**) PP50.

**Figure 5 polymers-12-00658-f005:**
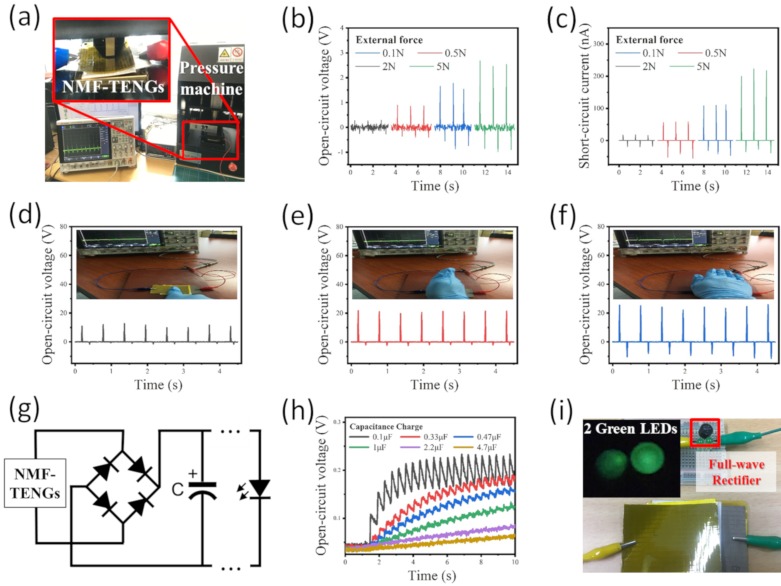
(**a**) Image of the measurement system. (**b**,**c**) Output voltage and current when applying external forces in the range from 0.5, 2, and 5 N. Output voltage with (**d**) finger, (**e**) blade, (**f**) palm tapping. (**g**) The bridge rectifier circuit. (**h**) Effect of capacitors on charging behavior. (**i**) Lightening green LEDs in the bridge rectifier circuit.

**Table 1 polymers-12-00658-t001:** Basic properties of the nonwoven PPs and nylon 6 nanofiber mat.

Label	Diameter	Thickness (μm)	Air Permeability (l/m^2^/s)	Mean Pore Size	Maximum Pore Size	Porosity (%)
PP15	2.4 ± 0.2 μm	180	766.3 ± 42.7	16.1	34.0	74.8
PP30	2.7 ± 0.2 μm	350	367.7 ± 17.2	17.9	33.8	77.3
PP50	2.7 ± 0.5 μm	600	266.0 ± 17.9	16.8	33.2	80.3
Nylon 6	283.8 ± 13.6 nm	80	86.6 ± 2.1	0.3	0.8	87.9

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
