# Peer review of "Nano- And Microfiber-Based Fully Fabric Triboelectric Nanogenerator For Wearable Devices"

_polymers, 2020, doi:10.3390/polym12030658_

Round 1

Reviewer 1 Report

In this study, the authors fabricated a nano- and micro-fiber-based TENG (NMF-TENG) through coupling a nylon 6 nanofiber mat and melt blown nonwoven polypropylene (PP) as triboelectric layers. A great deal of work about morphology, porosity, pore size distribution, and fiber diameters of the triboelectric layers and triboelectric output performances were investigated. However, the analysis of experiment dates and description of phenomenon in the article did not show all the advances of this paper. It would be a better work after deeper interpretation and understanding. The following questions will be helpful to deepen understanding.

  1. Compared with other work about fiber-based TENG, what’s the advantages of nonwoven PPs and nylon 66 used this article.
  2. Table 1 shows the air permeability and pore sizes of nonwoven PPs and nylon 6 nanofiber mat. Please elaborate on the effect of porosity and pore size on TENG output and give out
  3. The title is “Nano- and micro-fiber coupled fully fabric-based triboelectric nanogenerator for wearable devices”. However, the article does not discuss what’s the coupling effect of nano and micro fibers. Please give more explanations about the coupling effect.
  4. There are some mistakes in Figure 3c about the working mechanism of the triboelectric nanogenerators. Equal amount of opposite charge should remain stable in frictional layers during a work cycle. Please read the relevant literature (like, DOI:10.1039/c9tb02466b; doi.org/10.1038/s41467-019-10433-4; Advanced Materials, 2016, 28, 846–852) and make change to this figure.
  5. For broad readership of TENG and wearable devices, some literatures should be referred: Small, 2020, 1904758; InfoMat, 2020;2:212–234; Science Advances, 2016, 2, e1501478.
  6. The writing of the whole manuscript should be smoothed. There are lots of mistakes in grammar and spelling.

Author Response

To reviewer 1

Thank you for your kind and helpful comments. In response to your recommendations, we have revised the manuscript.

  1. We added recent references on fiber-based TENGs to the third paragraph of the introduction. In the additional references and lines 59 to 71 of page 2, we discussed the advantages and disadvantages of nanofibers and microfibers.

Triboelectric layers have been fabricated to maintain the advantages of nanofibers and microfibers, to compensate for the shortcomings, and to be easily applied to wearable devices. Therefore, we have fabricated NMF-TENGs with the triboelectric layer which is relatively-low cost, and offers easy control of the thickness as well as a large air volume and contact area.

  1. On page 6, we revised the manuscript (line 193) in response to your comments.

  1. The expressions “coupling” and “coupled” may be confusing to readers. The word “couple” simply implies that a triboelectric layer is fabricated by matching the nylon 6 nanofiber mat and nonwoven PP fabric. The reasons and advantages of using both nanofibers and microfibers are discussed on page 2. Furthermore, to avoid confusion among the readers, the expression “coupling” and “coupled” in the title and text were replaced.

  1. Figure 3(c) has been modified as per your suggestion on page 5.

  1. For further reading on TENG and wearable devices, we have added the reference you recommended to page 1 line 39.

  1. We have rechecked the manuscript for English grammar and spelling. To better communicate our findings and aid in easy understanding, the grammar and spelling in the manuscript were revised by “Editage - https://www.editage.co.kr”.

Again, we would like to thank you for your very useful comments.

Reviewer 2 Report

In the manuscript “Nano- and micro-fiber coupled fully fabric-based triboelectric nanogenerator for wearable devices”, the authors utilize nonwoven PP micro-fiber and Nylon electrospun fiber, which can be utilized as two contact layer materials. With the innate porous characteristics without additional post-treatment, the fabricated triboelectric nanogenerator can generate electrical output with its vertical contact-separation operation. The manuscript has novelty in its fabrication and structure but several aspects need to be addressed for further consideration.

  1. Is there any special reason why the authors perform the electrospinning the N6 nanofiber onto PP microfiber instead of directly onto Ni electrode? Would it be any difference between two processes? In my opinion, direct deposition of nanofiber onto the Ni electrode would generate higher output performance with more porous structures. Please compare the output performance between two cases.
  2. How about the electricity generation with sliding mode operation? As far as I understand, the present triboelectric nanogenerator can also easily generate output by using the sliding operation. Please check the output generation behavior with the sliding mode operation.
  3. How can the author measure the open circuit voltage with the oscilloscope? Please add comprehensive explanation on its measurement.
  4. The final goal of the developed triboelectric nanogenerator is to harvest the human related energy. Several relative recent papers in introduction may provide the new impact on enriching this work as the references.

    1) J. He, et al., “Piezoelectric-enhanced triboelectric nanogenerator fabric for biomechanical energy harvesting”, Nano Energy, 2019.
    2) S. Cho, et al., “Universal Biomechanical Energy Harvesting from Joint Movements Using a Direction-Switchable Triboelectric Nanogenerator”, Nano Energy, 2020.
    3) Z. Li, et al., “Multilayered fiber-based triboelectric nanogenerator with high performance for biomechanical energy harvesting”, Nano Energy, 2018.

Author Response

To reviewer 2

Thank you for your kind and helpful comments. In response to your recommendations, we have revised the manuscript.

  1. The final purpose of this study is to fabricate an integrated textile-type TENG device that can be effectively used for wearable electronics as clothing or footwear. Therefore, we fabricated the triboelectric layer using porous structural nonwoven without a spacer and operated by the full contact mode.
    As you recommended, we already performed the electrospinning of nylon 6 directly onto the Ni electrode as shown in Figure R1. However, it is necessary to add a bonding process between the nanofibers and the microfibers or coating process for the NMF-TENG device, as shown in Figure R3(b). In the case of the additional adhesive, it is difficult to generate the triboelectric effect as we expected (PP and nylon 6).

  2. As pe your suggestion, the friction through sliding mode operation between the microfiber layer attached to the electrode and the nanofiber layer fabricated by electrospinning nylon 6 directly onto the electrode clearly improves electrical performance better than operating the NMF-TENG fabricated in this study. However, continuous friction between the nylon 6 layer and the nonwoven PP layer causes the deformation of each layer surface like lint, as shown in Figure R4. Furthermore, our NMF-TENG devices were fabricated by electrospinning nylon 6 onto PP nonwoven, so they cannot operate in the sliding mode.

  3. Open-circuit voltage is the difference of electrical potential between two terminals of a device when disconnected from any circuit. There is no external load connected. The generator voltage was measured with an oscilloscope probe directly on the two electrodes of the NMF-TENG without load resistance.

  4. For better information transfer and reader understanding, we have added the references including your recommendations to the introduction.

Again, we would like to thank you for your very useful comments.
